Performance analysis of lightweight CNN models to segment infectious lung tissues of COVID-19 cases from tomographic images

http://orcid.org/0000-0002-0647-988X Iyer Tharun J. 1
http://orcid.org/0000-0003-1505-3159 Joseph Raj Alex Noel 2
Ghildiyal Sushil 1
Nersisson Ruban 1 nruban@vit.ac.in
1 School of Electrical Engineering, Vellore Institute of Technology , Vellore, Tamil Nadu , India
2 Department of Electronic Engineering, Shantou University , Shantou, Guangdong , China
Woźniak Marcin
Electronic publication date: 2021 Feb 22
Publication date: 2021
Volume: 7
Electronic Location ID: e368
Received 2020 Nov 3; Accepted 2021 Jan 3
Copyright: © 2021 Iyer et al.
Copyright year: 2021
Copyright holder: Iyer et al.
License: This is an open access article distributed under the terms of the Creative Commons Attribution License, which permits unrestricted use, distribution, reproduction and adaptation in any medium and for any purpose provided that it is properly attributed. For attribution, the original author(s), title, publication source (PeerJ Computer Science) and either DOI or URL of the article must be cited.
License URL: https://creativecommons.org/licenses/by/4.0/

Keywords: Convolutional neural networks, Computed tomography, COVID-19, Segmentation, High-Resolution Network (HR Net), Segmentation Network (Seg Net), UNet, VGG-UNet

Funding: The authors received no funding for this work.

==============================
The pandemic of Coronavirus Disease-19 (COVID-19) has spread around the world, causing an existential health crisis. Automated detection of COVID-19 infections in the lungs from Computed Tomography (CT) images offers huge potential in tackling the problem of slow detection and augments the conventional diagnostic procedures. However, segmenting COVID-19 from CT Scans is problematic, due to high variations in the types of infections and low contrast between healthy and infected tissues. While segmenting Lung CT Scans for COVID-19, fast and accurate results are required and furthermore, due to the pandemic, most of the research community has opted for various cloud based servers such as Google Colab, etc. to develop their algorithms. High accuracy can be achieved using Deep Networks but the prediction time would vary as the resources are shared amongst many thus requiring the need to compare different lightweight segmentation model. To address this issue, we aim to analyze the segmentation of COVID-19 using four Convolutional Neural Networks (CNN). The images in our dataset are preprocessed where the motion artifacts are removed. The four networks are UNet, Segmentation Network (Seg Net), High-Resolution Network (HR Net) and VGG UNet. Trained on our dataset of more than 3,000 images, HR Net was found to be the best performing network achieving an accuracy of 96.24% and a Dice score of 0.9127. The analysis shows that lightweight CNN models perform better than other neural net models when to segment infectious tissue due to COVID-19 from CT slices.

Introduction

During the winter months December 2019, a highly contagious disease out broke in Wuhan, China (Zhu et al., 2020; Liu et al., 2020). High grade fever and other flu like symptoms were noticed, and most of the patients developed pneumonia. The pathogen causing the disease was identified as corona virus, and named as Severe Acute Respiratory Syndrome Corona Virus-2 (SARS-CoV-2) (World Health Organization, 2020). The disease caused by the virus is named by World health Organization (WHO) as Corona Virus Disease (COVID-19). WHO also declared COVID 19 spread as Global public health emergency (World Health Organization, 2020). As of 9th May 2020, more than 200 countries around the world are affected by COVID-19. There are around 4 million people affected by the disease worldwide with a mortality rate of 6% (around 2.75 million people lost their lives). Developed countries like US, Europe and most of the developing countries are suffering a lot from the outbreak. The scientific community is largely involved in devising an antidrug and vaccines for the device. But unfortunately there are no positive results till date and more over it is reported that, due to mutations, characteristics are changing which makes the vaccine development even more challenging. Taking the situation into account, there are very few ways we can control the virus; like staying isolated from the world and breaking the spreading chain of the virus, maintaining the personal hygiene, early detection of the symptoms and taking necessary precautions are few of them.

The successful control of the outbreak depends on the rapid and accurate detection and identification of the symptoms isolating the patient from the community, so that the spread of the disease can be stopped. Currently, the method used for the detection is Real-time reverse transcriptase polymerase chain reaction (RT-PCR) (Li & Xia, 2020). It is the standard procedure used by many hospitals and clinics for testing COVID-19 cases. Even though this method remains the reference standard, there are many reported false negative cases using this RT-PCR (Chan et al., 2020), which is an alarming fact on the situation. It is also time consuming and the limited supply of RT-PCR kits for the rural areas make the testing more difficult (Chen, Yao & Zhang, 2020). Since the COVID-19 patients develop breathe related discomfort and Pneumonia as the outcome of the disease progress, Radiological studies can play a vital role in diagnosing the lung infections caused by this episode (Zu et al., 2020). The CT chest scan can be used to identify the early stages of lung infections and related problems. The chest CT reveals the initial pulmonary abnormalities for COVID-19 patients for whom RT-PCR gave negative results (Ai et al., 2020).

Also, to accurately and efficiently control the virus, studies have been conducted to implement forecasting models to predict the spread of COVID-19 (Wieczorek, Siłka & Woźniak, 2020). Due to the nature of the problem being a regression problem to forecast the spread and predict how the virus may spread, Artificial Neural Networks (ANNs) and Recurrent Neural Networks (RNNs) were used to model the data. Data was collected from the center for Systems Science and Engineering (CSSE) at Johns Hopkins University. To further improve the accuracy of prediction and decrease the error rate, Deep Learning was widely used as a predictor and forecasting model. Generative Adversarial Networks (GANs), Extreme Learning Machine (ELM), and Long/Short Term Memory (LSTM) were some models used to predict the spread of the virus (Jamshidi et al., 2020). The performance of the Deep Learning differed significantly from the use of RNNs and ANNs. Therefore, we decided to use Deep Learning methods in our study to explore the performance of models on our data. To approach the solution of using Deep Learning methods to segment lung CT scans, we looked at studies conducted on detecting small nodules or lung tissue using complex neural networks. Capizzi et al. (2019) used probabilistic neural networks and a Bio-Inspired Reinforcement Learning network based on fuzzy logic to accurately segment lung nodules. The network worked at 92.55% accuracy and considerably lowered the computational demands of the detection and segmentation system. Ke et al. (2019) proposed a neuro-heuristic algorithm to segment lung diseases from X-ray Images. The algorithm achieved an average accuracy of 79.06% to segment and classifies three diseases in the X-ray Images. But for our study, CT Scans were chosen as the primary source of data due to the easy availability and higher number of slices per scan. Therefore, we would have more data than X-ray Images.

The common manifestations of SARS-CoV-2 in chest CT scan are ground glass opacities, consolidation, crazy paving, dilation of vessel width in some cases and round shape lesions in few cases (Hamer et al., 2020; Zu et al., 2020). The effectiveness of the chest CT scan based COVID-19 management depends on the efficient automatic detection and segmentation of regions in the scan. So in that context, the recent developments in the imaging technologies come handy. There are plenty of imaging tools which give very high and accurate quantification of abnormal conditions. This procedure of image based diagnosis system involves capturing the image, analyzing the image by a trained, experienced radiologist and annotation is made for the ground truth segments. The current scenario slows down the annotation of the images, labeling and getting the ground truth processes due to the increasing number of patients day by day, lack of radiologist and the over duty burden of existing radiologists. Therefore, automatically detecting the infected regions from the chest CT scan using computer based algorithms are the current trends in research that gives wonderful results and aids in medical diagnostics.

The main objective of the research work is, comparing the segmentation performance of computationally non-intensive models deep learning model when subjected to Lung CT Scans for that are affected by COVID-19. The models utilized for the research belong to the U-Net variants models which are the most popular models of choice for segmentation of Medical Images. Here we compare the traditional U-Net model as proposed by Ronneberger with other variants such as Seg Net, U-Net based on VGG16 and High Resolution Net (HR-Net) and present both qualitative and quantitative results.

Materials and Methods

The block diagram describing the entire methodology is shown below in Fig. 2. The description of each method is described below.

Figure 1 Comparison between original image (A) and pre-processed image (B).

Dataset considered

The used dataset consists of 3,770 images and their corresponding ground truths. A total of 3,020 training images are used and 750 testing images are used. The CT scans of 50 patients were taken from mosmed.ai (Morozov et al., 2020) were openly accessible Neuroimaging Informatics Technology Initiative (NIFTI) images were provided. The data was collected from the Research and Practical Clinical Center for Diagnostics and Telemedicine Technologies of the Moscow Health Care Department. The CT Scans were obtained between 1st March 2020 and 25th April 2020. Each NIFTI file was decompressed to PNG images and used for the study. The CT scans of another 20 patients were taken from zenodo.org (Jun et al., 2020) where the NIFTI files of 20 patients were provided. The images were annotated by two radiologists and verified by an experienced radiologist. For both datasets, MATLAB was used to extract the PNG images.

Pre-processing

The images in the dataset are riddled with motion artifacts and noise. Motion artifacts are caused due to improper imaging techniques and are a specific kind of noise relevant to CT Scans. Therefore, removing this noise is important or else it will cause the algorithms to learn improperly. MATLAB is used to remove the noise and motion artifacts. The original image is converted to grayscale from RGB and then, the image properties are extracted. Area and Solidity are used and then, the image is thresholded after selecting the max area and highest solidity. Once a mask is ready, the mask is multiplied with the original image to get the pre-processed image. A comparison is given below in Fig. 1. As can be seen, motion artifacts are removed and the image has more clarity. Other pre-processing methods to remove the noise and motion artifacts from Lung CT Images are using a mean filter (Khan, 2019) and a series of region growing and morphological applications (Devarapalli, Kalluri & Dondeti, 2019). These methods were mainly used to remove the sharp edges in the CT scans and to smoothen the image so that the network could learn better. But, on comparison of the different methods, our pre-processing method provided a better performance in all metrics

Figure 2 Block diagram of proposed method.

HR Net

HRNet is developed at Microsoft and has signified state of art presentation in the areas of semantic segmentation, image classification, facial detection, object detection and pose estimation (Sun et al., 2019). Its attention is on training High Resolution (HR) representation. The existing techniques recuperate representation of high resolution from representation of low resolution formed by high to low resolution network. In HRNet, from first stage commencement high-resolution network, progressively augment high to low resolution networks successively to arrange more steps and associate the multi-resolution network in parallel.

HRNet is able of uphold high-resolution representation throughout the process as repeated multi-scale combinations are conducted by switching the information through the multi-resolution parallel subnetworks repeatedly throughout the process (Sun et al., 2019). The architecture of resulting network is displayed in Fig. 3. This network has advantages in contrast to existing networks like Segnet, UNET, Hourglass etc. These existing networks lose a lot of essential information in the progression of recovering high-resolution from low-resolution representation. HRNet links high to low resolution networks in parallel instead of series and this gives high-resolution representation throughout the process, correspondingly the estimated heatmap is much accurate, spatially much precise.

Figure 3 Architecture of HRNet.

Multi-resolution sequential subnetwork

Existing models works by linking high to low resolution convolutions subnetwork in series, where each individual subnetwork form a platform, collection of an arrangement of convolutions furthermore, there is a down sample layer through end-to-end subnetworks to split the resolution into halves.

Let Nsr be the subnet in the stage sth and resolution index r. First subnet resolution is given by 12r−1. The high-to-low system with S phases/stages (i.e., 4) can be indicated as: (1) N11→N22→N33→N44

Multi-resolution parallel subnetwork

Starting from first phase/stage begin with high resolution subnet, slowly enhance high to low resolution subnet, generating new phases/stages, and associate multi-resolution subnet in parallel. Eventually, the parallel subnet resolution of a later phase/stage comprises of the resolution from an earlier stage and below one stage. The network shown below contains 4 parallel subnets.

Multi-scale repeated fusion

In this network exchange units were introduced throughout parallel subnet in such a way that an individual subnet continuously collects information from parallel subnets. How information is exchanged lets understand this process through an example here third stage is subdivided into multiple exchange blocks and every block consists of three parallel convolution modules, having exchange units followed by parallel units which is shown below:

where:

Csrb – Convolution module,

εsb – Exchange Unit,

and s is the stage, r is the resolution and b is the block

Explanation of exchange units is show in Fig. 4. The input mapping is given by: {X1,X2,X3,…,Xs} and the output mapping was given by: {Y1,Y2,Y3,…,Ys}. The width and resolution of the output is same as input. Every output is a sum of input mapping that is, YK=∑i=1s⁡a(Xi,K). Assume of 3×3 stride was done for down sampling and for up sampling 1×1 convolution (nearest neighbor).

Figure 4 Layers of exchange unit of convolutions with various resolutions: (A) low, (B) medium and (C) high resolution.

HRNet experimental results (when tested with different datasets) show remarkable results for the applications like facial detection, semantic segmentation, and object detection.

Seg Net

At the university of Cambridge, UK, team of the robotics group researched and developed that SegNet is a deep encoder decoder architecture for multiclass pixel-wise segmentation (Badrinarayanan, Kendall & Cipolla, 2017). The framework comprises order of non-linear processing layers which is called encoders and a similar set of decoders afterward a pixel wise classifier. Generally, encoder have made up of a ReLU non-linearity and one or more convolutional layers with batch normalization, subsequently non-overlapping maxpooling and subsampling. Using Max-pooling indices in encoding sequence, for up sampling the sparse encoding in consequence the pooling process to the decoder. Use of max-pooling indices in the decoders is the one important feature of the SegNet to execute the sampling of low resolution maps. For segmented images the tendency to retain high frequency details and capable enough to decrease the number of parameters in the decoder needed for training are some advantages of SegNet. Using stochastic gradient descent this framework can be trained end-to-end.

SegNet is composed of encoder and decoder after a last pixel-wise classification layer. The architecture is shown in Fig. 5. The encoder in SegNet is composed of convolution layers which are 13 in number, and these layer matches with the 13 starting layers of VGG16, considered for classifying the objects (Mannem, Ca & Ghosh, 2019).

Figure 5 SegNet architecture.

Photograph source credit: Google Earth image, ©2015 Google.

Figure 6 illustrates the decoding method utilized by SegNet in which there is no learning engaged with the up-sampling stage. The upsampling of decoder network’s feature map (input) is done by learned maxpooling indices from the equivalent encoder feature map. Dense feature maps are generated by combining feature maps and trainable decoder channel.

Figure 6 Decoding techniques used by SegNet.

SegNet a deep network was used for semantic segmentation. Basically, It was designed because the motivation behindhand was to propose an architecture for roads, outdoor and indoor sites which is proficient together in terms of computational time and memory. Feature map’s maxpooling indices are only stored in SegNet and to attain better performance it uses them in its decoder network.

UNet

The UNet design is based upon the fully convolution network and adjusted such that it produces better segmentation results in medical imaging. UNet consists of two paths named as contracting and and expansive. In the contracting path it captures the context whereas in expansive path it enables exact localization. While contracting path is a classical architecture of UNet. It includes two 3 × 3 convolutions, max pooling operation with repeating application. The Fig. 7 illustrates the architecture of UNet, which is U in shape that itself gives the name “UNet”. The main philosophy behind this network is, it replace pooling operation by using upsampling operators (Ronneberger, Fischer & Brox, 2015). So, ultimately the resolution will increase layer by layer. The main feature of UNet is the large number of channels which lead to higher resolution. Moreover, in every downsampling it doubles the feature channels.

Figure 7 Illustrating the architecture of U-Net.

Each stage in the expansive path involves upsampling of the feature channel followed by (2 × 2) convolution that splits the number of feature channels into halves. In contracting path, it crops the feature map because of loss in border pixel in each convolution. Final layer is mapped by 1 × 1 convolutions which is used to map all 64 units feature vector. The network contains total 23 convolutional layers. UNet performs well on image segmentation (Livne et al., 2019).

While training the UNet model, the cross-entropy loss function united with the last feature map and by applying a pixel-wise softmax over it, the softmax is denoted as: (4) pk=e(ak(x))∑k′=1K⁡e(ak′x)

In addition, the energy function is calculated by: (5) E=∑x∈Ω⁡w(x)log⁡(pl(x)(x))

where:

ak: Represents the activation in feature map k

pk: Represents estimated maximum function

K: No. of class

x∈Ω: Pixel position

pl(x): Deviation

In the training data set, to counterbalance the diverse frequency of pixels from a specific class the weight map is pre calculated for ground-truth segmentation, and enforcing the network to study the minor separation borders amid touching cells introduce by us.

The morphological operation used to calculate separation borders, the weight map calculated using: (6) w(x)=wc(x)+w0.e(−(d1(x)+d2(x))22σ2)

where:

w: denotes the weight map

d1: distance upto border of nearest first cell

d2: distance upto border of nearest second cell

VGG UNet

Image segmentation, which is performed pixel wise is most preferable task in the field of computer vision. Encoders-decoders when combined they form UNET architectures, which are very famous for image segmentation in medical imaging and satellite images etc. The weights of the pre-trained models (like ImageNet) are used to initialize the weights of the neural network (i.e., trained on large dataset) as it gives better performance major than those models, which are trained on small dataset from scratch. Models accuracy is very important in some applications like traffic safety and medicine pre-trained encoder can enhance the architecture and performance of UNET. Applications like Object detection, image classification and scene understanding have improved their performance after the introduction of convolutional neural network (CNN). Nowadays, CNN has outperformed in several fields over human experts.

Image segmentation plays vital role in the field of medical imaging to enhance the diagnostic capabilities. Fully connected network (FCN) is amongst the most popular state-of-the-art machine learning technique (Long, Shelhamer & Darrell, 2015). Segmentation accuracy attained by some advancement in FCN as compared to PASCAL VOC (Everingham et al., 2015) common approach on standard datasets UNet consists of two paths named as contracting and and expansive. In the contracting path it captures the context whereas in expansive path it enables exact localization. The contracting path sticks with the design of a convolutional network with pooling operations, alternating convolution, gradually down sample feature channels and expanding many feature maps layer simultaneously, each stage in the expansive path composed of an up-sampling of the feature channel along with a convolution. The VGGUnet architecture is illustrated in Fig. 8. The encoder for UNET model is composed of 11 successive (series) layers VGG family and denoted by VGG-11 (Ai et al., 2020). VGG-11 consist of 7 convolution layers each using rectified linear unit (ReLu) activation function, 5 maxpooling operations each reduces feature channel by 2 and the kernels size 3 × 3 is used for every convolutional layer (Iglovikov & Shvets, 2018).

Figure 8 Illustrating encoder decoder architecture also known as VGGUNet.

Common loss function that is, binary cross entropy can be used for classification problem where y^i denotes the prediction, yi denotes the true value and m denotes the no. of samples (7) H=−1m∑i=1m⁡(yilog⁡y^i+(1−yi)log⁡(1−y^i))

Performance validation

To validate the performance of the models presented above, Sensitivity, Specificity, Jaccard Index, Dice Coefficient, Accuracy and Precision are used. To measure the accuracy of the segmented image, accuracy and precision are used and to measure the quality of segmentation, sensitivity and specificity are used. The various performance measures are described below:

Accuracy and Precision are used to calculate the accuracy of the segmentation model itself. Accuracy is as the ratio of correct predictions to the total number of predictions and Precision is defined as the ratio of correctly predicted positive observations to the total number of correctly predicted observations.

(8) Accuracy=TP+TNTP+TN+FP+FN

(9) Precision=TPTP+FP

In the case of segmentation, accuracy and precision are used to measure the binary segmentation of each pixel of the image by the model. Although precision and accuracy may seem to be enough to describe the performance of the model, other factors are also important to describe the quality of segmentation.

Sensitivity and Specificity are used to measure the quality of segmentation between the classes. In this case, the models are performing binary segmentation. So, Sensitivity, or the True Positive Rate, measures the quality of segmentation of one class and Specificity, or the True Negative Rate, measures the quality of segmentation of the other class. Sensitivity and Specificity can be defined as: (10) Sensitivity=TP(TP+FN)

(11) Specificity=TN(TN+FP)

With Sensitivity and Specificity, having a high value for each is good as it shows that the model is able to segment the pixels correctly without any errors.

The Jaccard Index and Dice Coefficient are used to quantify the similarity between the original image and the segmented image. Jaccard Index and Dice Coefficient are similar to the Intersection over Union (IoU) used to evaluate Object detection models. Jaccard Index and Dice Coefficient ranges from 0 to 1 where 0 means no overlap and 1 mean full similarity. Jaccard Index and Dice Coefficient can be defined as: (12) DiceCoefficient=2TP2TP+FP+FN

(13) JaccardIndex=TPTP+FP+FN

While the Dice coefficient and the Jaccard Index are quite similar, ideally, the two measures have to be equal. So, to measure the quality of similarity of the model, a similarity between the Dice Coefficient and Jaccard Index can be viewed to measure the quality of segmentation.

Results and Discussion

The experiments were conducted in the Google Colab platform. As shown in Table 1, HR Net is shown to have the highest performance as compared to the other models. The second best model is the classical UNet, the third best model is the VGG UNet and the model with the worst performance is the Seg Net. The reason for the high performance of the HR Net is the fact that the HR Net extracts high resolution information and retains in throughout the segmentation process. This is due to the parallel networks that are able to maintain essential information. HR Net indicates a high accuracy of segmentation with an Accuracy of 0.9624 and a Specificity of 0.9930. The performance is also compared against heavy weight models that have more parameters and layers than the lightweight models. The weight size is also considerably larger. As per our study, lightweight models offer better performance as compared to heavy weight models in all evaluation metrics. Especially in Dice Coefficient, Accuracy and Precision, the lightweight models like HR Net and UNet offer better performance than Inception ResNetV2 and ResNet 101. SegNet and VGG UNet are comparable to the performance of the heavy weight models.

Table 1 Performance measures of segmentation performed on four light weight and two heavy weight models mentioned.

Method	FPrate	FNrate	TPrate =
Sensitivity	TNrate =
Specificity	Jaccard	Dice	Accuracy	Precision	
HR_Net	0.0142	0.1085	0.8862	0.9930	0.8428	0.9147	0.9624	0.9593	
Seg_Net	0.2148	0.2141	0.7859	0.7952	0.7962	0.8014	0.8816	0.8416	
UNet	0.1260	0.1785	0.8215	0.9195	0.8143	0.8836	0.9105	0.9281	
VGG_UNet	0.2259	0.2082	0.7918	0.7964	0.8224	0.8418	0.8794	0.8416	
Inception ResNet V2	0.1879	0.1762	0.8064	0.8267	0.8069	0.8154	0.9268	0.9154	
ResNet 101	0.2567	0.2481	0.8341	0.8249	0.8395	0.8254	0.9087	0.9354	

Figures 9 and 10 shows the various outputs obtained from the models which segmented the positive tested and negative tested image respectively. HR Net shows the best segmentation performance while UNet shows good performance too. UNet is able to obtain a proper boundary similar to the test image. Seg Net has performed poorly to segment the image. Neither has it obtained a boundary nor has it segmented the finer details properly. The VGG UNet has segmented the image properly but not to the extent of HR Net or UNet. We can see that HR Net has the best performance. With decreased area, the performance decreases which means that UNet is the second best performance, VGG UNet is the third best and Seg Net has the worst performance amongst the models. When we compare Figs. 9 and 10 with Tables 1 and 2, we can review the performance of the models on COVID positive and negative slices. On comparing HR Net with ResNet 101 and InceptionResNetV2, we can see that HR Net shows comparable performance to the heavy weight models but still performs better in terms of performance metrics like Jaccard Index and Dice Coefficient. This disparity is especially seen in terms of Accuracy. The Specificity of HR Net is also much higher than the heavy weight models. This can be attributed to the method in which HR Net extracts the features. Even though the numbers of layers are more, it still retains a smaller size than the heavy weight models without sacrificing on performance. The next best performer is UNet as can be seen from the images. UNet is able to segment the boundaries and consistencies of the slices but cannot maintain the shape in all the predictions. VGG UNet and Seg Net perform the worst each with having their own disadvantages. VGG UNet might be better at detecting finer details in the lungs but cannot maintain the basic predictive ability to detect boundaries and textures. Seg Net performs the worst as it cannot segment even the most basic boundary or details. Since it is used more as a segmentation algorithm for land masses, this makes sense for Seg Net to perform badly in the case of Lung CT slices.

Figure 9 COVID positive tested with all the four lightweight and two heavy weight models.

(A) Original image and predictions from (B) UNet (C) Seg Net (D) VGG UNet (E) HR Net (F) ResNet 101 and (G) Inception ResNetV2.

Figure 10 COVID negative tested with all the four lightweight and two heavy weight models.

(A) Original image and predictions from (B) UNet (C) Seg Net (D) VGG UNet (E) HR Net (F) ResNet 101 and (G) Inception ResNetV2.

Table 2 Inference speed of the four light weight and two heavy weight models along with other parameters which affects the inference speed.

Name of model	Inference time (ms)	Number of layers	Number of params	Model size
(MB)	
UNet	42	32	7,759,521	30	
SegNet	84	64	31,819,649	122	
VGG UNet	65	85	25,882,433	99	
HR Net	140	1043	28,607,456	112	
Inception ResNet V2	123	572	55,873,736	215	
ResNet 101	115	101	44,675,560	171	

Figure 11 is the glyph plot which is a visual representation of the performance metrics for each model. The glyph plot is a good way to directly compare the performance of various models through the use of polygons. We can see that HR Net has the best performance. The six points are the performance measures namely Sensitivity, Specificity, Precision, Accuracy, Jaccard Index and Dice Coefficient. With decreased area, the performance decreases which means that UNet is the second best performance, VGG UNet is the third best and Seg Net has the worst performance amongst the models.

Figure 11 Glyph plot representing the performance of the models over the performance measures.

(A) HR Net; (B) Seg Net; (C) UNet; (D) VGG UNet (The 6 points are the performance measures namely Sensitivity, Specificity, Precision, Accuracy, Jaccard Index and Dice Coe).

As can be seen from Table 2, HR-Net has the highest number of layers at 1,043 with other models having less than 100 layers. But, HR-Net has similar parameters to VGG UNet and an even lesser number of parameters than SegNet. This is due to the architecture of the HR-Net. It is able to extract deeper features than the other models while maintaining the overall file size and number of parameters. The reason for HR-Net having the highest performance is its architecture which makes it the best model to be used for fast inference. Comparing the SegNet, VGG UNet and UNet, SegNet has the poorest inference speed at 84 ms with the largest model size. It is therefore inferred that SegNet is the worst model to use for the Segmentation of COVID-19 based on our study. VGG UNet and UNet have different metrics due to the fact that VGG UNet is trained on the VGG-16 weights. It, therefore, takes far higher time to load on and produce an inference than UNet. If HR-Net cannot be used, VGG UNet is the next best network for segmenting COVID-19. To check whether the performance of lightweight models is better than previous literature, we have checked the performance with “heavy-weight” models. Heavy weight models can be classified as those models with more number of layers; parameters and weight file size in Mega Bytes (MB). The Heavy weight models do not offer better performance than the lightweight models in terms of inference time. The heavy weight models are slower than the lightweight models in segmenting COVID-19 from CT images and also, take up more memory space. While the inference time is faster than HR Net, the number of parameters and model size does not allow the models to perform as well as the light weight models.

Conclusion

In this article, we analyzed four models for segmenting COVID-19 from Lung CT Images. With the growing number of cases worldwide, quick and accurate testing is needed. To solve this problem, we approached the problem by reviewing four lightweight models that do not take a long time for training or testing. First, we remove motion artifacts from the tomographic images through Thresholding and use the pre-processed images for training the models. The four models trained are Seg Net, UNet, VGG UNet and HR-Net. We evaluate the models on their performance using accuracy, dice, Jaccard index and precision. We also used Specificity and Sensitivity as secondary evaluation characteristics. The results obtained demonstrate that lightweight convolutional networks have high latent ability to segment COVID-19 from CT images with HRNet being the best network out of the four models analyzed. Our work can be used in real-time environments to deploy on low-power devices. Low-Power devices require less computation time and have many constraints. When we consider these constraints, then using our lightweight models is very efficient as the user can accurately segment COVID-19 from CT Images. This system can be used in the field where electricity is constrained and fast and accurate predictions are required. The proposed light weight model can be implemented a simpler hardware that requires less area and power requirements. Due to the lower power usage, the prototype can be used as standalone systems in power constrained conditions but require accurate predictions. The results in this study can be improved by collecting more data from hospitals and clinics to improve the accuracy of the segmentation. We can also improve the work by changing the architecture of the proposed models to extract more features without increasing the inference time significantly.

We would like to thank VIT University for the nurturing environment as well as the exposure it provided for us to complete this project. We also would like to thank MosMedData for the Covid 19 dataset which is used for this research work.

Additional Information and Declarations

Competing Interests

Author Contributions

Data Availability

The authors declare that they have no competing interests.

Tharun J. Iyer performed the experiments, performed the computation work, prepared figures and/or tables, and approved the final draft.

Alex Noel Joseph Raj conceived and designed the experiments, authored or reviewed drafts of the paper, and approved the final draft.

Sushil Ghildiyal performed the experiments, performed the computation work, authored or reviewed drafts of the paper, and approved the final draft.

Ruban Nersisson analyzed the data, authored or reviewed drafts of the paper, and approved the final draft.

The following information was supplied regarding data availability:

Code and data are available at GitHub: https://github.com/IyerOnFyer/COVID-19-Segmentation.git.

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
