# Peer review of "Performance analysis of lightweight CNN models to segment infectious lung tissues of COVID-19 cases from tomographic images"

_PeerJ Computer Science, doi:10.7717/peerj-cs.368_

## Round 0.1 · original submission · Major Revisions

Dear Authors, Reviewers suggest changes to your paper by solving important issues. Please revise it and return to us with revisions for further processing.

Reviewer 1 ·

Basic reporting

This article presents an in-depth analysis of the performence of selected, most-known Deep Convolutional Neural Networks, such as SegNet or VGG UNet, on prepared dataset containing lungs' CT slices of patients with COVID-19.
Authors have prepared a well written and understandable descriptions of used CNN models giving the recipient complete knowledge from some basic fundamentals to more specific characteristics.
The experimental setup is clearly described.
There are however some areas to improve.

Experimental design

Authors of this article claims that the lightweight CNN models perform better than other NN models. However according to the Table 2 we can clearly see that the best results are achived by the HR Net which, in comparison to other 3 NN's from this article, is the most complex in terms of the number of hidden layers and it is the second highest model in terms of Model Size.
If however all this models are classified by the authors as light-weight, there should be a comparison between a deeper model and the selected once to justify the claim of this article.

Validity of the findings

Findings are well described and does not contain any major errors.

Additional comments

The article is written with mostly correct English, with some minor drawbacks (For example in section "Dataset Considered" there is written "The dataset used consists of..." where there should be "The used dataset consists of...").
Also some places should be corrected visually (to be more consistant), for example in Introduction there is "scan are:- ground glass opacities,consolidation,...". The colon implies that it is a list of elements, but there is also a single dash only before the first element. It makes the text look inconsistant and less professional.

I would also suggest adding references to articles below because of similar topics and interesting applications of deep learning in COVID-19 treatment:
- "Neural Network powered COVID-19 spread forecasting model"
- "Artificial intelligence and COVID-19: deep learning approaches for diagnosis and treatment"

Reviewer 2 ·

Basic reporting

Improve the quality/resolution of image in equation (3).

Improve the quality of equation (4) (6) which are unreadable, make quality and resolution higher.

Improve the quality of some pictures (fig 2, fig 5, fig 6, fig. 10).

Check if the article meets the PeerJ style requirements.

Experimental design

More explain the pre-processing operation. It would be interesting if you use other pre-processing method and compared them.

Explain obtained results presented on the fig 9.

On the Figure 10 are presented glyph plots for used methods. In my opinion it's not necessary but in other way you must increase the quality of the graph. I think moreover you should analyze and discuss the data from table 1.

Validity of the findings

The article is interesting but need some extensions and modification. Can you show how your proposition work correctly and incorrectly in case of some input images?

Additional comments

Do you have any other example where your method can be used? Do you think that your proposed method can be improved and how? Authors can extend the bibliography to include articles: Neural network powered COVID-19 spread forecasting model, Small Lung Nodules Detection based on Fuzzy-Logic and Probabilistic Neural Network with Bio-inspired Reinforcement Learning, A neuro-heuristic approach for recognition of lung diseases from X-ray images

---

## Round 0.2 · accepted · Accept

Dear Authors,

The review process is now finished. All Reviewers are positive about your manuscript therefore I am pleased to forward your manuscript with a positive recommendation.

Reviewer 1 ·

Basic reporting

Article presents well documented analysis of the selected models on prepared dataset containing lungs' CT slices of patients with COVID-19

Experimental design

Authors have responded clearly on given concerns. No further actions are required.

Validity of the findings

Findings are well described and there are no visible mistakes.

Additional comments

Authors have responded clearly to all given previously concerns. The article is clear and well written.

Reviewer 2 ·

Basic reporting

I have no critical comments.

Experimental design

I have no critical comments.

Validity of the findings

I have no objections.

Additional comments

The authors followed my suggestions and answered my questions clearly, so I have no further objection.